# NDIS Participants with Psychosocial Disabilities and Life-Limiting Diagnoses: A Scoping Review

**DOI:** 10.3390/ijerph191610144

**Published:** 2022-08-16

**Authors:** Kathy Boschen, Caroline Phelan, Sharon Lawn

**Affiliations:** 1College of Medicine and Public Health, Flinders University, Adelaide, SA 5050, Australia; 2College of Nursing and Health Sciences, Flinders University, Adelaide, SA 5050, Australia

**Keywords:** National Disability Insurance Scheme (NDIS), psychosocial disability, severe and persistent mental illness, life-limiting, palliative care

## Abstract

This research aimed to map evidence about system supports and gaps for Australians with psychosocial disabilities and life-limiting diagnoses. A scoping review of available policy documents, academic, and grey literature was completed to discover key characteristics of this concept and provide context around the phenomenon. Our focus was on Australia’s National Disability Insurance Scheme (NDIS), a key reform providing support to the disability population nationally. No peer-reviewed or grey literature was retrieved on the phenomena. Therefore, three lines of enquiry were developed: experiences of NDIS participants living with psychosocial disabilities; the death, dying, and palliative care supports and experiences of NDIS participants of any disability type; and the experiences for people living with severe and persistent mental illness (SPMI) and life-limiting diagnoses. Five themes were identified: (1) the person; (2) advocacy; (3) informal supports; (4) formal supports; and (5) existing research. NDIS participants living with SPMI and their informal and formal support systems are still struggling to navigate the NDIS. While there are no specific publications about their end-of-life experiences, people with SPMI often experience poor end-of-life outcomes. Rigorous research into their death, dying, and palliative care experiences is needed to inform improved support to them, including their end-of-life care.

## 1. Introduction

Introduced in 2013, the National Disability Insurance Scheme (NDIS) is a world-first, once-in-a-generation reform to the Australian disability sector [1]. To ensure the sustainability of the NDIS, Australian Governments have agreed that where services are best funded under another service system, the NDIS will not fund those supports. However, there is an expectation that the NDIS will support ‘participants’ in the scheme seamlessly across service systems:


*The interactions of people with disability with the NDIS and other service systems should be as seamless as possible, where integrated planning and coordinated supports, referrals and transitions are promoted, supported by a no wrong door approach.*
[2] (p1)

By 2030, the NDIS will support an estimated 859,328 participants, and approximately 88,180 (10.3%) will enter the scheme with a primary psychosocial disability (predominantly related to severe and persistent mental illness) [3]. There appears to have been little consideration about how formal support systems intend to support NDIS participants that receive life-limiting diagnoses [4]. It is essential to understand whether the current service systems, NDIS, health, and mental health around these NDIS participants adequately support them once they receive a life-limiting diagnosis. People with severe and persistent mental illnesses often die up to 20 years sooner than the average [5,6,7,8,9,10,11,12,13]. However, as around 80% of people who die in Australia are over 65 years, most palliative care programmes and support appear to be geared towards the over-65s. Further, there is a paucity of research into the death, dying, and palliative care experiences of people living with severe and persistent mental illness [5,14,15,16,17,18,19,20,21,22,23,24]. This limited research also shows that people with psychosocial disabilities experience stigmatising, siloed, disjointed systems, and substantial support gaps. Therefore, these issues are amplified when they are palliating, and their end-of-life (EOL) experiences are typically bleak and inequitable [16,17,21,25,26,27,28,29]. This scoping review seeks to discover existing publications regarding NDIS participants with psychosocial disabilities and life-limiting illnesses and identify concepts and the characteristics of any systems gaps to inform future research.

## 2. Methods

NDIS March 2022 data reveals that participant numbers are sitting at ~60% [30] of the total estimated number of participants. Consequently, comprehensive quality research regarding a full-scheme NDIS is limited. Therefore, the most appropriate way to understand the available information is a scoping review to map the available evidence about the phenomena of interest to identify gaps in knowledge [31]. Using Arksey and O’Malley’s framework, a scoping review of the available policy documents and academic and grey literature was undertaken to discover key characteristics of this concept and provide context around the phenomenon [32,33]. The Scoping Review-Prisma-ScR Checklist can be found at Appendix B.

### 2.1. Development of Key Lines of Enquiry

Search strategies were developed with the assistance of a research librarian, and database searches were subsequently performed within the CINAHL, Scopus, Medline via Ovid, and Psychinfo databases. Further review of the National Disability Insurance Agency (NDIA), NDIS Quality and Safeguards Commission (NDIS Commission), Australasian Legal Information Institute (AustLII), and Department of Social Services and Australian university websites were undertaken to search for relevant web pages, guidelines, and government position statements. The database search resulted in no peer-reviewed studies, published opinions, grey literature, or policy documents regarding NDIS participants with a primary psychosocial disability who have been diagnosed with a life-limiting illness. Three key lines of enquiry were subsequently derived from the research question to review the literature from the following perspectives:What information is available regarding the NDIS experiences of participants living with psychosocial disabilities?What information is available regarding the death, dying, and palliative care supports and experiences of NDIS participants?What information is available regarding the death, dying, and palliative care experiences of people living with psychosocial disabilities?

### 2.2. Publication Search Considerations

A subsequent search, using the three key lines of inquiry, was completed 14 March 2021 using CINAHL, Scopus, Medline via Ovid, and Psychinfo. The authors developed the following inclusion and exclusion criteria.

Exclusions: Searches of the databases and websites used the three identified themes and excluded publications before 2013 (prior to the NDIS’s commencement). The Australian Government enacted the National Disability Insurance Scheme in 2013, and the Australian Palliative Care Standards 5th Edition and the Fifth National Mental Health and Suicide Prevention Plan were also released that year. Thus, 2013 represents contemporaneous paradigm shifts in the disability, mental health, and palliative care sectors that reflect current practice and perspectives. As part of the establishment of NDIS, the Council of Australian Governments (COAG) agreed that all early intervention supports for mental health conditions are best funded by mental health. While early intervention is not limited to people under 18, most people under 18 should be receiving early intervention supports through the mental health system [2]. Therefore, all publications regarding under 18-year-olds were excluded from the study. Publications not in English, and publications that were epidemiology or health promotion-focused, were also excluded.

Inclusions: The term “serious and persistent mental illness” was used in the palliative care publications to enable the sector to identify that patients/clients had a mental illness that was not secondary to their life-limiting diagnoses. Search terms for each database and theme are detailed in Appendix A.

### 2.3. Publication Selection

Initial searches resulted in 5701 papers; after removing duplicate publications, 5061 were identified for title and abstract screening, and 272 for full-text screening. Two authors (K.B. and C.P.) independently reviewed included publications, supported by the Covidence “Better systematic review management” document 2018 (available from: https://www.covidence.org/home (accessed 14 March 2021)). All conflicts were resolved without the need for adjudication. Additional texts were included through citation tracking and new publication alerts. The scoping literature review contains 49 publications under the NDIS and psychosocial disability line of enquiry; 66 publications under the SPMI, death, dying, and palliative care line of enquiry; and two publications under the NDIS and death, dying, and palliative care line of enquiry. A total of 117 publications were therefore included. Publication selection flow chart diagrams, based on the Prisma Report for each line of enquiry, can be found in Figure 1, Figure 2 and Figure 3.

## 3. Results

The NVivo software tool was used to code and sort the literature. Common themes and connections were identified and analysed against the three lines of enquiry that were derived from the initial scoping review question:What information is available regarding the NDIS experiences of participants living with psychosocial disabilities?What information is available regarding the death, dying, and palliative care supports and experiences of NDIS participants?What information is available regarding the death, dying, and palliative care experiences of people living with psychosocial disabilities?

Across the literature, common themes were discovered regardless of the literature’s service system, issue, or intent. Data were, therefore, coded into these common themes. Five key themes and a range of sub-themes within three of the themes were developed as described in Figure 4:

### 3.1. The Person

#### 3.1.1. People with Psychosocial Disabilities and the NDIS

Psychosocial disabilities were a late addition to the NDIS [34], and the scheme’s impact on the lives of people living with psychosocial disabilities is yet to be fully understood. To access the NDIS, a person living with a psychosocial disability needs to meet the disability requirements found in Section 24 of the National Disability Insurance Scheme Act (2013) [35]. Mental illness is one of the leading causes of disability nationally and internationally [19,36]. However, there has been significant, cogent discourse in the mental health community about the appropriateness of the NDIS for Australians living with mental illness. Most notably, it was found that there has been further fragmentation of the support systems experienced by people with SPMI [37].

#### 3.1.2. People with Serious and Persistent Mental Illness and Palliative Care

The term “psychosocial disability” is not commonly used in the palliative care literature. Instead, the terms “severe and persistent mental illness” (SPMI) and “severe mental illness” (SMI) identify patients/clients with significant mental illnesses that are pre-existing and not secondary to their life-limiting diagnoses [17,26,38,39,40,41].

People living with SPMI have increased somatic risks, higher cancer mortality rates [42], poorer clinical outcomes [43], and die significantly earlier due to factors attributable to their mental illnesses. These factors include the side effects of psychotropic medications, unhealthy lifestyles [38], alcohol and other drug use, poor health monitoring [44,45,46], and reduced health prevention and screening, leading to under-detection and late diagnosis of disease [18,22,25,26,39,40,45,46,47,48]. People with SPMI are particularly vulnerable to shorter life expectancy [40,49] as they are often victims of violence [50], healthcare system neglect [25,45], and can be excluded from mainstream service support due to barriers such as homelessness [26,27,51], cultural insensitivity [52], poverty, and stigmatisation [5,14,15,17,18,21,22,24,25,26,27,39,47,49,51,53,54,55].

#### 3.1.3. Stigma and End-of-Life Care for People with SPMI

Fear of discrimination due to past experiences of stigma often leads people with SPMI to disconnect from services and supports [49], resulting in unmet needs across a range of areas, including mental and physical healthcare, housing, and alcohol and other drugs (AOD) treatment. In addition, this fear of discrimination can lead to ambivalence regarding receiving treatment and end-of-life and palliative care [14,21,22,26,39,51,56]. There is, however, across the research ample confirmation that people with SPMI are almost universally stigmatised within health and palliative care settings, resulting in substantial inequity, unmet need, reduced access to care, and poor end-of-life outcomes [5,14,15,18,21,22,24,25,26,27,28,39,49,51,53,54,57,58,59].

#### 3.1.4. Psychosocial Disabilities, Capacity and Decision-Making

A frequent discussion about people with both disabilities and SPMI within the palliative care, mental health, and NDIS literature [52,60,61,62] was their cognitive capacity and ability to make decisions about their supports, palliative care, and end-of-life care. Many of the studies discussed the presumptions within medical and mental health settings that people with SPMI were, due to capacity issues, unable to make decisions or that the symptoms of their mental illness made discussions about death, dying, and palliative care overwhelming [5,15,16,17,18,36,38,39,40,56]. Promoting and respecting existing relationships, such as with carers and multidisciplinary health/mental health teams, is key to ensuring that people with psychosocial disabilities are well supported as they die [5,15,16,22,24,63]. Where capacity exists, people with SPMI have the same rights as others to make poor decisions [40]. Notably, the NDIS Practice Standards uphold this concept of, and right to, dignity of risk [64].

#### 3.1.5. Human Rights of People Living with Psychosocial Disabilities and Life-Limiting Illnesses

Multiple studies reveal that healthcare for people with SPMI is not equitable [14,15,16,17,19,21,25,26,39,45,65,66,67,68,69]. Ethical challenges such as withheld treatments due to SPMIs and concerns about risks to other patients [67], fewer referrals and admissions to palliative and quality end-of-life care [14,15,16,25,45,68], and being subjected to more invasive end-of-life treatments, such as intubation, CPR, and feeding tubes [47], and being denied access to mental health care teams in their healthcare setting [16] were described. The United Nations Convention on the Rights of Persons with Disabilities’ (UNCRPD) purpose is to “promote, protect and ensure the full and equal enjoyment of all human rights and fundamental freedoms” and to “promote respect for their inherent dignity of people with disabilities” [70] (p. 4). The Objects and Principles of the NDIS Act give effect to the UNCRPD [35]. However, concerns regarding the human rights of people with disabilities [50,71,72], and/or SPMI, living with life-limiting diagnoses are expressed frequently throughout the literature [14,15,27,38,39,73]. Grassi and Riba [39] state that dignity is incompatible with stigma. The stigma of mental illness that has been pervasive throughout people’s lifetimes is compounded by intrinsic and extrinsic factors when managing significant health conditions [26].

Healthcare supports also have additional ethical complications when people with SPMI decline or withdraw from treatment [25]. Concerns regarding patient capacity, vulnerability, and risk often override their right to healthcare choices and advance care planning, and result in them being subjected to involuntary treatments [25].

Unfortunately, the NDIS has not been the promised panacea for people with disabilities since its introduction. Instead, Australia has seen an uptick in guardianship applications and financial management orders [60]. Additionally, carers of people with disabilities report that the NDIA/Local Area Coordinators (LAC) staff have inadequate levels of understanding of disability and do not have the requisite empathy and compassion that would ameliorate this inexperience [74].

This scoping review found that the human rights of people living with SPMI and a life-limiting illness are not being upheld, particularly the rights of equity, freedom of discrimination, dignity, the right to housing, and equitable access to healthcare.

#### 3.1.6. Insecure Housing as a Barrier to Palliative Care

As a signatory to the UNCRPD, Australia recognises the right to and, therefore, should ensure [70] that all people with disabilities have access to public housing. However, the responsibility of housing sits within each state and territory governments’ jurisdiction. The NDIS, therefore, does not ensure that NDIS participants are guaranteed housing if their support needs do not meet the threshold for supported disability accommodation [2]. Globally, people with SPMI are at a high risk of homelessness, and housing insecurity is recognised across the literature a significant barrier to palliative care [14,16,18,26,27,39,40,51,53]. Where people are not street living, they may be living in hostels, supported residential facilities, mental health facilities or in shared accommodation settings that make the delivery of in-home palliative care unrealisable or challenging [18].

#### 3.1.7. Palliative Care and Human Rights

Quality palliative care is a recognised human right [75,76]. However, the literature identified that people with disabilities and SPMI often experience inequity and have significant unmet needs [25,27,29,49] and barriers to palliative care [14,17,18,27,40,47,49,68].

#### 3.1.8. Discussing Death, Dying and Palliative Care

Formal mental health and palliative care support providers expressed significant concern regarding discussing death and dying with people with SPMI; they also expressed concern that this would exacerbate the symptoms of their mental illness [18,22,27]. However, when researchers conducted interviews with people living with SPMI, the concerns and themes were inconsistent with clinician concerns [22]. Indeed, in one study, people with SPMI advised that they were aware of this avoidance by clinicians, increasing their sense of abandonment [22]. Another study found that, contrary to clinician beliefs that their research participants did not experience fear of death, many people with SPMI found it a relief to talk about and that themes of death had been regularly contemplated throughout their lives [36]. Skilled companionship at the end of their lives was identified as crucial to improving end-of-life experiences for people with SPMI [36]. Additionally, the loss of their providers of mental health supports, who many view as “de facto family” is the source of significant distress [49,53,77]. People with SPMI highlighted the importance of their formal supports being trained, in some form, in both mental health and palliative care, ensuring people with SPMI and life-limiting conditions are treated holistically and do not have their care compartmentalised or avoided due to clinician discomfort [22,36,53].

#### 3.1.9. Quality of Life

“Improving quality of life…as well as reducing physical and mental stress” [78] (p. 4) is fundamental to palliative care. People with SPMI often have a reduced quality of life throughout their lifetime, further compounded by a life-limiting diagnosis [25,36,79]. Many people with SPMI have struggled with service systems that do not respect their autonomy. When diagnosed with life-limiting conditions, there can be further declines in agency and physical capabilities with increased dependence on or being a burden to loved ones or others [36,40].

#### 3.1.10. Symptom Management

There is often a misattribution of symptoms, diagnostic overshadowing occurs, and care teams believe physical symptoms are due to a patient’s mental illness due to their communication style [16,26,36,51,73]. However, a study by Jerwood et al. [19] found that people with SPMI may hide their symptoms due to the difficulties of building new therapeutic relationships with new care teams. Communication issues with and underlying stigmatising beliefs held by health professionals can also lead to under-prescribing pain medications for people with SPMI at the end of their lives [16,25,26,53]. Quality of life during palliative and end-of-life care can be supported by managing pain [78], and research has identified this as a significant issue for people with SPMI [36,40,53,80]. Unfortunately, there appears to be confusion within the literature regarding perceptions of pain for people with SPMI. Many publications refer to a decreased response to pain [15,38] or an inability to talk about their pain in a way that others can understand [36,51,73].

### 3.2. Advocacy

NDIS participants and their informal supports require significant system knowledge and negotiation capabilities to navigate the NDIS. A lack of confidence or capacity to understand the NDIS can result in participants not receiving the support they need [60]. Carey, Malbon and Blackwell [52] advise that personalisation systems such as the NDIS require prodigious amounts of self-advocacy. Participants must understand their support needs and figure out how the NDIS can meet them [81]. Participants must know how to set NDIS goals in support plans and demonstrate that supports for that goal are not best funded by another support system, and communicate their goals and connect their support needs, and NDIS supports, to those goals [82]. Additionally, Malbon and Blackwell found that paid advocacy results in increased funding within NDIS plans [52]. A lack of advocacy is highlighted as a contributing factor to poor outcomes for people with psychosocial disabilities in the NDIS and palliative care systems. The acknowledged importance, yet underfunding of advocacy organisations in Australia [83], would further contribute to this issue.

### 3.3. Informal Supports

#### 3.3.1. Limited Informal Supports

Many people living with SPMI have little to no informal support; this can be family or friends that undertake an unpaid carer role throughout their lives. This loss can add further complexity when they are diagnosed with a severe medical condition or life-limiting illness [14,16,18,22,26,28,38,39,40,47,49,68,73,84,85,86], and attempting to navigate systems on their own [14]. Advocacy is essential to respect end-of-life advance care directives when a person has no informal supports [18,49]. Substitute decision-makers are often state-funded guardians, and there can be limitations on their powers. Decisions may be required from tribunals or courts, causing unacceptable delays in care [26,87]. Psychiatric nurses describe being considered substitute family members [5], and mental health teams often have close relationships with their clients [86]. However, people with SPMI are likely to be separated from these essential supports when they enter the health system due to their life-limiting illness [15].

#### 3.3.2. Lack of Respect for Informal Supports

Informal supports or carers describe a lack of respect by formal support services when supporting a person with SPMI and a life-limiting condition [22,88]. Health and mental healthcare providers expect them to display expertise in the medical and psychiatric conditions of the person they are caring for, to be their advocate, and attend to their personal and intimate care needs [22,24], even though informal supports are often perceived as problematic by formal supports [5,18,22]. Informal supports safeguard people with SPMI from the gaps between silos [77]. However, they describe having to keep extensive records and being ignored when they had concerns about the person they were caring for; only heard if they expressed that they could no longer cope [89].

#### 3.3.3. NDIS and Informal Supports

The shift to the NDIS has caused a decline in support for informal carers of people living with mental illness in Australia [34,90]. Informal supports describe extensive difficulties negotiating with NDIA [34]. LACs and planners have been described as judgmental, dismissive, and possessing limited capabilities to work collaboratively with informal supports of NDIS participants [74]. As part of the NDIS process, carers are often excluded from planning meetings, resulting in unmet participant support needs. Supports and service funding once used to support carer well-being have been redirected into the NDIS; however, NDIS funds, for the most part, provide support to NDIS participants [34], not informal supports. Other government carer supports are minimal [34], and searches through the Carer Gateway, a government website dedicated to supporting carers, provide no clear answers about carer support. Attempting to navigate these systems substantially strains relationships between the NDIS participants and their informal and formal supports [34].

#### 3.3.4. Investing in Informal Supports

The emotional and physical burdens and social isolation experienced by informal supports is significant [88], and it is important to acknowledge their significant economic contribution. Diminic et al.’s [90] research into the caring hours provided by informal supports of people living with SPMI in Australia estimated that they provide 186 million hours of unpaid work each year. Without these informal supports, people with SPMI would need to have more of their support needs funded by the NDIS, including the high-cost supported independent living (SIL) or independent living options (ILO). If these hours were to be funded by the NDIS, at the minimum 2021 hourly support rate of AUD 57.10, the cost of replacing informal supports would be around AUD 10.62 billion. These figures do not count any weekend, public holiday, afternoon and night shift loading, or rural or remote loading per the NDIS pricing arrangements [91]. Indeed, the “Mind the Gap” report estimated the cost of replacing unpaid carer hours at AUD 13.2 billion in 2018 and advised that not investing in unpaid carers would severely impact the funding and delivery of social services in Australia [34].

### 3.4. Formal Supports

There is limited research into the formal support systems of NDIS participants with psychosocial disabilities and NDIS participants with any disability who are dying, and none regarding the formal support of NDIS participants with a psychosocial disability who are dying. There has also been a dearth of research regarding how organisations can best support people with SPMI who receive a life-limiting diagnosis. However, this limited research reveals that formal support systems are inadequate [5,16,18,22,54] despite the multiple service providers involved in the lives of people with SPMI [14,17,51]. Furthermore, even though mental health, NDIS, and palliative care providers share person-centred values [15,19,40,92,93], collaboration between providers, though highly recommended and encouraged, is poor, limited or non-existent [15,16,18,21,22,27,38,40,53,92,93,94,95,96,97,98,99]. The NDIS does not fund case management or care coordination, a recognised and highly valued role in mental health systems [100,101], to the detriment of NDIS participants with complex support needs. Support coordination, specialist support coordination, and LAC roles do not have clear guidelines on how to provide support, nor the funding nor jurisdiction to provide this complex support [94,102]. The NDIA does briefly explain the role of specialist support coordinators on its website; however, the criteria to obtain funding for specialist coordination in an NDIS plan are not provided. However, many NDIS participants and their informal supports are uncertain of what is available and how to request particular support [52]. There is no publicly available data on how many NDIS participants are receiving specialist support coordination or how many hours are funded. Isaacs and Firdous’s [96] review of the now defunded Partners in Recovery program demonstrated that care coordination was cost-effective and efficient in supporting people living with SPMI while maintaining recovery-orientated practice. While the NDIA has co-opted recovery terminology, there is little evidence or capacity for recovery-oriented practice within the NDIS, due to its deficits-based approach [103,104,105].

Traditionally, mental health systems case-managed people with SPMI; however, resource limitations often no longer provide this comprehensive case management and are usually restricted to monitoring medication and compliance [104]. For people with SPMI who are dying, lack of care coordination [18,22,27,40] and appropriate standardised tools [15,38,53] are barriers to palliative care and contribute to poor end-of-life experience. General practitioners find navigating the NDIS [106] and palliative care systems [49] challenging, and they may have little time to spend with [49] or be responsive to the needs of [89] their patients with SPMI. The complex care needs of people with SPMI with life-limiting conditions, and a lack of appropriate referrals to specialist palliative care, results in unmet needs, distressing end-of-life outcomes, and the inability to develop trust, a therapeutic alliance, and advance care directives [18,22,27,49,53,68,86]. Workers in the palliative care [14,107], mental health [18], and NDIS sectors [50] need to ‘bend the rules’, work unpaid hours and go above and beyond their system’s funding to support people with SPMI throughout their lives and as they die [16,25,27,38,61,94,108]. The literature describes concerns about the risks to and safety of staff and other patients that can negatively impact the delivery of palliative and end-of-life care to people with SPMI [5,14,18,22,26,39,109], sometimes leading to the need for restrictive practices such as chemical or physical restraint [40,109]. There are, at this time, no publications or guidelines available regarding practices restrictive of people with disabilities that discuss the requirements of the NDIS Commission combined with the requirements of the various state health systems or My Aged Care. Further, no available publications or operational guidelines discuss supporting NDIS participants with psychosocial disabilities to die at home, whether in their own home or supported disability accommodation.

#### 3.4.1. Mental Health

In Australia, mental health systems consist of government-run/funded mental health systems and non-government organisations that may receive funding from either federal or state programs. The reallocation of resources to the NDIS has reduced funding in the community mental health systems around the country to varying degrees [34]. Secondary losses include loss of qualified staff and rural and remote programs, a casualised workforce, and hybrid providers that provide NDIS and fund other mental health supports, resulting in streamlining and loss of supports offered [37].

These losses of qualified supports are concerning, given that people with SPMI and life-limiting illnesses often lose access to their mental health services due to being absorbed into the health systems [53]. As a result, they can be discharged from mental health services without notice or have supports reduced. The literature shows that this leaves people with SPMI and life-limiting conditions feeling abandoned and dying alone in unfamiliar environments [22,79]. The literature also highlights the need for continuity of care and continued support from a multidisciplinary team where pre-existing therapeutic relationships are maintained and fostered, and medication management responsibilities are shared [22,24,53,79,94]. However, the research indicates that the mental health workforce often finds working with dying patients/clients challenging [5,16,28]. In addition, many organisational guidelines are not conducive to palliative care in mental health settings [18].

Mental health services have difficulties supporting dying clients [5,15,18,27,40,49,53,109,110,111,112]. This may be due to an inability to provide high-level somatic care, funding models or operational guidelines that see people with SPMI discharged from mental health supports when they enter the health system [18]. Mental health system staffs advise that they are apprehensive about supporting their clients as they die, as the skills required are not within their standard care practice and discussions about death can be confronting [18]. For some people with SPMI in Australia, mental health facilities and SDAs are considered their homes, and dying in familiar surroundings is important for some people with SPMI [36]. People with SPMI have advised that they have not been provided with information about palliative care by their mental health teams even as they withdrew their support [22].

#### 3.4.2. NDIS

Similar to trends in human services in some European countries, the Australian Government designed the NDIS to deliver individualised or self-directed support to people with disabilities [95,113]. These funding models help empower NDIS participants [114] and ratify Australia’s obligations under the UNCRPD [35]. However, the implementation of the NDIS has spawned many issues since it commenced transition in 2013. The NDIS is built on a foundation of middle-class norms that may lead to high administrative burdens and poorer outcomes for those from more marginalised communities [52]. David and West [71] advise a lack of “nuanced empirical data about the long-term effects of marketisation in the disability sector” (p. 333). They suggest that market-driven approaches to social services may be regressive. Cortis and van Toon [115] expressed concern regarding the private market and self-regulation of providers, and “loose parameters of oversight” (p. 122).

The NDIA has struggled to effectively support NDIS participants with psychosocial disabilities [1,34]. The literature identifies issues such as the inability to plan around fluctuating conditions, and the dichotomy of the disability model and deficits-based language with the recovery model and its associated terminology used by the mental health systems [34,37]. Systemic power imbalances cultivated within the NDIA continue through to the service delivery landscape. NDIS workers are paid less in the NDIS system than in other systems [34,105]. Many support workers have few or no qualifications, particularly in mental health, and are subjected to insecure work arrangements. There are few opportunities for professional development, and many feel they will not continue working in the disability sector. This, ultimately, will limit choice and control, further disempowering NDIS participants, particularly those in rural and remote areas [34,37,50,74,82,105,116,117,118,119].

Providers are reporting that they must work around the NDIS rules to survive financially [34]. The conflict between quality and profit has negatively impacted their organisations’ missions, making NDIS participant relationships transactional [34,95]. Due to insecurity around income, the financial risks to businesses have caused NDIS providers to restructure their business models to ensure financial viability. As NDIS participants can change NDIS providers with limited notice, they are curating the types of disability supports offered to reduce these financial risks [95,117].

NDIS providers report that the caps and lower remuneration from the NDIS have resulted in difficulties with recruiting and retaining qualified staff [74,92,117], particularly those with mental health training [34,37]. They advise that the needs of NDIS participants already exceed the system’s ability to supply the supports required, particularly in rural and remote areas [74,95]. In addition, thin market issues have not been alleviated by the new, inexperienced NDIS providers entering the sector [95]. Competition is also impacting NDIS providers’ collaboration: while Foster et al. report that providers reducing the types of services offered has increased collaboration [99], competition often negatively impacts interagency cooperation [94,95]. Even though the NDIS stresses the importance of collaboration between NDIS providers and other service systems, no funding, policy or legislation frameworks support this [94].

#### 3.4.3. Palliative Care

Support for palliative care remains the health system’s responsibility [1,2]; however, it is essential to clarify how the various systems interpret palliative care, illness, and disability. For example, there are several genetic conditions on List A, the NDIA’s list of conditions that are likely to meet the disability requirements in Section 24 of the NDIS Act [35] that are life-limiting [120]. However, no published framework or guideline describes how the NDIA determines whether the palliative care system or NDIS best funds a support need [4]. The 2021–2031 National Disability Strategy policy priority [121] advises that people with disabilities should be able to choose where they live. As dying at home is the choice of many Australians and including people with SPMI [18], it is crucial to understand how the NDIS and the healthcare system intend to fund an NDIS participant’s choice and control when they are dying.

While there are no publications regarding providing palliative care support to NDIS participants with primary psychosocial disabilities, palliative care providers globally experience difficulties supporting people with SPMI [5,15,18,27,40,49,53,73,77,109,110,111,122]. Many people with SPMI are not receiving palliative care and often present to the hospital in the final stages of their life-limiting conditions [14,15,17,39,53]. They may die in acute care settings without receiving palliative care support [14,39,69]. This may be due to problems with identifying their illness [14,15,38] or that just surviving each day, and attempting to meet their basic needs of food and shelter lowers the priority of caring for their health [14]. Subsequently, they may only present for medical assistance when their symptoms become unbearable [14,39].

The siloed nature of mental health, health, and palliative care systems has been highlighted as a barrier to palliative care for people with SPMI [5,14,15,17,18,21,27,38,39,49,109]. Often there are no ongoing relationships with medical teams or distrust of medical professionals [40,49]. GPs can be an excellent support for this cohort; however, this is not consistent, and they may also act as a barrier to palliative care [27]. Due to funding allocations, a limited number of patients are admitted into specialist palliative care, and few facilities can manage the complexity of patients with SPMI [5,15,18,111]. The diagnosis of SPMI itself may be a barrier to referral to palliative care [17,39]. The literature demonstrates that palliative care is usually structured to support normative populations [14,38]. Subsequently, clinicians struggle to support people with SPMI adequately within existing systems.

Medical professionals may struggle with diagnostic overshadowing and prescribe fewer pain medications to people with SPMI [16,25,26,123] either due to stigma toward them by health professionals [26,39], or communication issues [51,73] and misattribution of symptoms [51]. Multiple publications indicate that palliative care patients with SPMI experience less pain or communicate pain differently to other people [15,26,38,51]. However, Jerwood et al. [22] and Sweers et al. [36] advise that people with SPMI may not be experiencing less pain. The inequity of palliative care leads to distress and poor symptom management for people with SPMI; this can also cause long-lasting harm to informal supports and friends [21].

People with SPMI are discharged from palliative care settings due to their SPMI symptoms impacting somatic care [25,39,40,68] and risk to other patients and staff [5,14,27]. In a similar challenge to clinicians from the mental health sector, palliative care clinicians may feel uncomfortable supporting people with a pre-existing mental illness and may feel alarmed by the active symptoms of an SPMI [39]. In addition, without knowing a patient with SPMI’s usual presentation or treatment regimen, it can be challenging to support them adequately [49,53] or manage medication interactions effectively [15,19,47,51]. Further challenges can include that some patients with SPMI can be ambivalent about dying or their end-of-life care [14,51], refuse treatment [25], and there may be uncertainty about the patient’s capacity for advance care planning [16,17,18]. However, studies have indicated that people with SPMI often have that capacity [27,38,80] and appreciate flexible, supported decision-making to develop their advance care directives [16,18,22,36,108,124].

#### 3.4.4. Training

The literature identified that people with a life-limiting diagnosis and a pre-existing SPMI benefit from a cohesive multidisciplinary team to support them as they die [25,39,49,53]. Therefore, best practice would be that mental health teams continue to support people with SPMI once they enter other systems [109,125]. However, both mental health and palliative care clinicians recognise that they lack the necessary skills and require additional, targeted training opportunities to ensure they can confidently provide competent and caring support for this complex cohort [5,16,18,21,27,38,40,49,53,77,111,126].

#### 3.4.5. System Design Issues

Each state and territory government entered into bilateral agreements with the Commonwealth Government regarding how the NDIS would be funded and delivered in each state and territory. The bilateral agreements have been amended at various stages of the implementation of the NDIS, and each state and territory has negotiated slightly different arrangements with the Commonwealth (detail can be found on the NDIA website [127]). In 2015, the COAG (now National Cabinet) developed the NDIS Principles to determine the responsibilities of the NDIS and other service systems [2]. However, while these set out funding models and responsibilities, there is no clear framework or pathways where these systems intersect [94,119]. Historically the state and federal governments have been responsible for different systems. The recognised issue with gaps between siloed systems [4,5,14,17,18,21,22,26,27,38,39,49,53,60,68,77,89,92,94,96,98,128,129] has been exacerbated, rather than resolved, with the addition of the NDIS [49,94,102,119].

The design of the NDIS, while attempting to focus on the individual, has resulted in “Taylorist administration” leading to “routinisation and data-driven planning” [60]. Carey et al.’s [52] scoping review found that the NDIS has been designed and works best for middle-class, white people. It is administratively burdensome and difficult to navigate [52,130] and excludes or provides less support for NDIS participants who are not from this social class. Neoliberal approaches to human services reduce the ability of people with disabilities and providers to work together to advocate for improved support [103]. Hummell et al.’s [94] rapid review found that while the design and intent of the NDIS was to increase collaboration between systems, there has been a reduction in information sharing and collaboration due to increasing competition. There needs to be a significant cultural shift to change the administration and policy directions of the NDIS [52] to overcome gaps in the NDIS and health system frameworks.

### 3.5. Existing Research

Despite the significant issues raised within the literature about the death, dying, and palliative care experiences of people with SPMI, there has been limited research nationally and internationally [5,14,16,17,18,19,20,21,22,23,24,26,40,109,131]. There is recognition that the perspective of the person with comorbid SPMI and life-limiting diagnosis is largely missing from existing research. It is acknowledged that it is necessary to conduct further research that includes their and their informal supports’ perspectives [18,22,36,132]. Research into the NDIS is usually directed toward difficulties regarding access and obtaining funded supports and workforce and NDIS provider issues. There has been limited research where NDIS participants have been offered the opportunity to contribute to our understanding of the NDIS [37] and none about the death, dying, and palliative care experiences of NDIS participants with primary psychosocial disabilities [37].

## 4. Conclusions

This scoping review revealed that there has been no research into the death, dying, and palliative care experiences of NDIS participants with psychosocial disabilities and life-limiting diagnoses. Furthermore, there was minimal literature across the three key lines of inquiry regarding experiences of NDIS participants living with psychosocial disabilities, the death, dying, and palliative care supports and experiences of NDIS participants of any disability type, and the death, dying, and palliative care experiences of people living with severe and persistent mental illness psychosocial disabilities. The experiences of NDIS participants are still highly variable, and the impact on NDIS participants with psychosocial disability and their workforce across the NDIS and other service systems has been significant. Additionally, this limited information demonstrated that while there are no publications about their end-of-life experiences, people with SPMI often experience poor end-of-life outcomes. The scoping review also discovered that the three lines of enquiry, training, cross-training, and mapping across the service systems are recommended to improve NDIS participant service experiences and EOL support. This mapping would be ideal for all NDIS participants where there may be intersections with other services. Another key finding is the importance of investment to strengthen and sustain informal support networks and advocacy services to ensure that NDIS participants can be well supported at the end of life in both the NDIS and other service systems.

Hamilton et al. [37] identified in their scoping review of the NDIS and psychosocial disability that there is a need for independent and rigorous research into the NDIS. It is essential to consider that while 88,180 participants are expected to enter the NDIS with primary psychosocial disabilities, many NDIS participants may enter the scheme with secondary psychosocial disabilities. Data regarding the number of NDIS participants with declared secondary psychosocial disabilities have not been made publicly available through the NDIS Data website. Therefore, it is difficult to understand the impact of co-occurring disabilities on EOL experiences. Further research must also extend to the death, dying, and palliative care experiences of NDIS participants of any disability type and consider the formal support they will receive from the NDIS and the other services systems. The NDIA must support research into this phenomenon to ensure that, in line with the goals of palliative care, the suffering of NDIS participants and their families is relieved and that they experience the best possible quality of life and equity in death.

## Figures and Tables

**Figure 1 ijerph-19-10144-f001:**
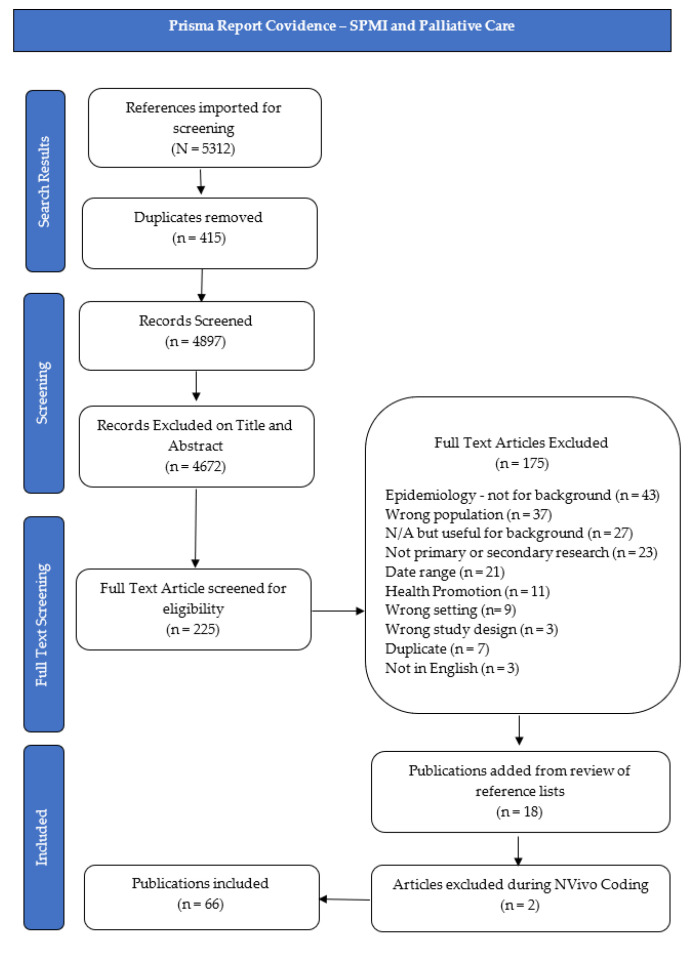
Prisma Report Covidence—SPMI and Palliative Care.

**Figure 2 ijerph-19-10144-f002:**
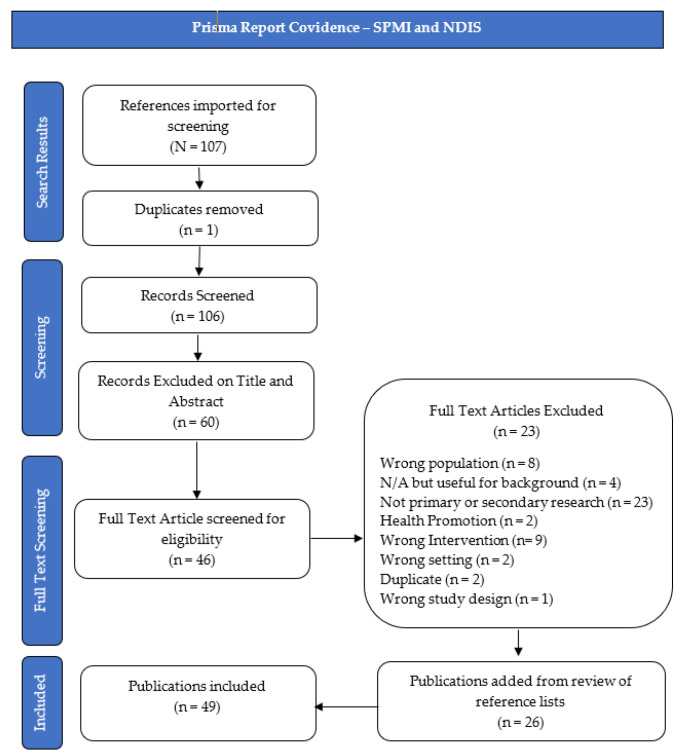
Prisma Report Covidence—SPMI and NDIS.

**Figure 3 ijerph-19-10144-f003:**
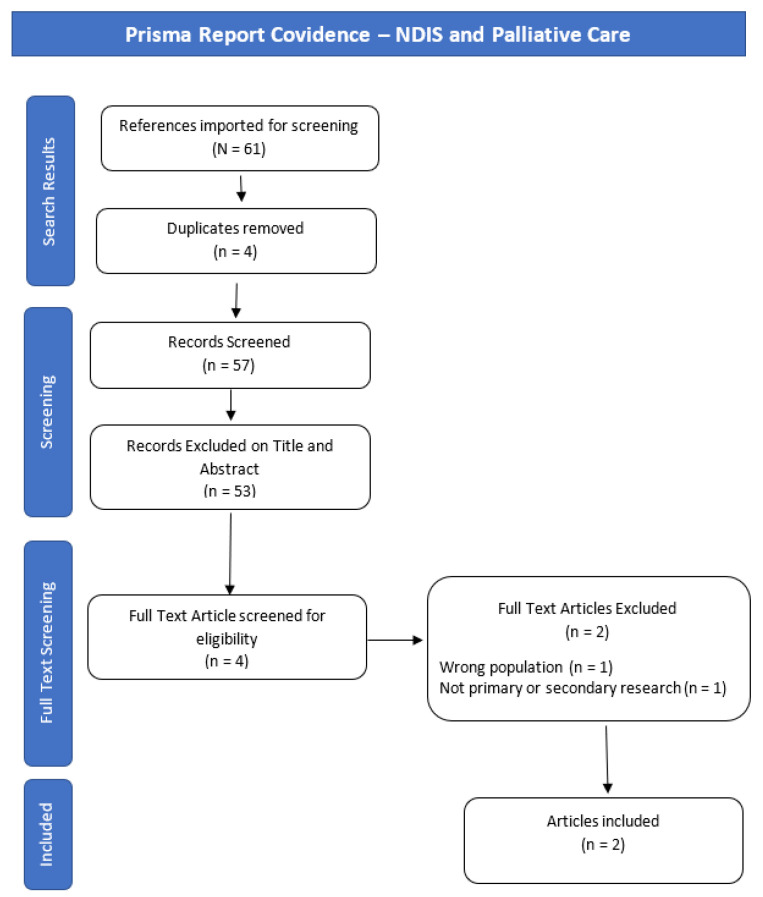
Prisma Report Covidence—NDIS and Palliative Care.

**Figure 4 ijerph-19-10144-f004:**
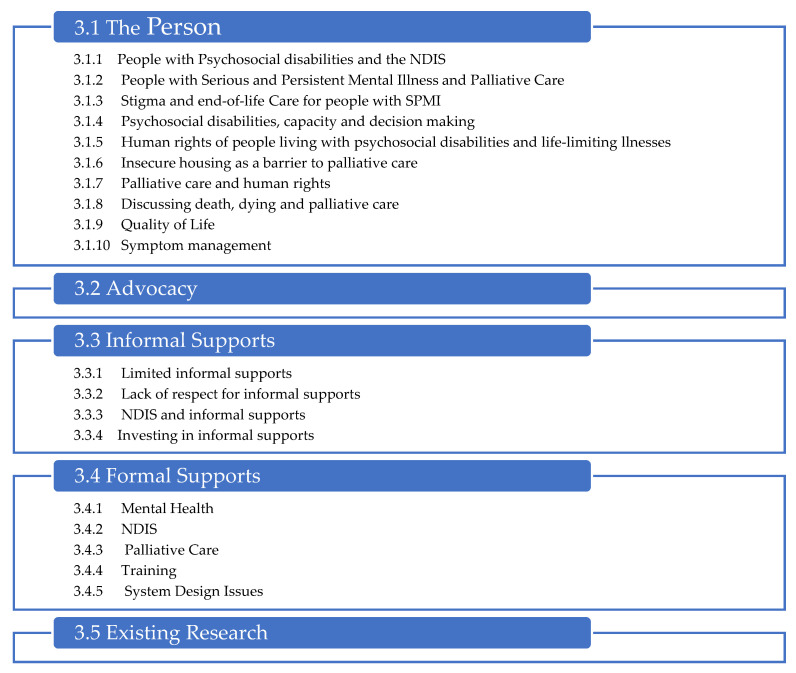
Scoping Review—Themes.

## Data Availability

Not applicable.

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
