# Peer review of "NDIS Participants with Psychosocial Disabilities and Life-Limiting Diagnoses: A Scoping Review"

_ijerph, 2022, doi:10.3390/ijerph191610144_

Round 1

Reviewer 1 Report

This paper focuses on the system supports gaps for Australians with psychosocial disabilities and life-limiting diagnoses.  This is a relatively small number in terms of NDIS participants, but an important one particularly given the inequities noted in the paper. The paper draws attention to the myriad of challenges for individuals who require support from not just the NDIS but a range of other mainstream services, which play out in a number of ways for different groups.  I think this is an important paper and worth publication subject to including more detail on the approach to coding of the data and also pulling together a discussion section. 

It is true that there is limited consideration given to psychosocial disability and a relatively small proportion of people within the scheme with psychosocial disability. But there is also likely a much greater number of people within the scheme who have psychosocial disability as a secondary disability. It might be good to see some consideration given to this issue.  

The methodology is mostly solid but there is no information provided regarding how the papers were coded or analysed and this needs to be added to the publication before it can be published. 

I agree that advocacy is an important issue for many NDIS participants.  While support coordination is available for all participants, a number of studies have demonstrated that for some groups they need specialist support coordination delivered by individuals who clearly understand the issues facing individuals.  It appears to me that this is such a group given the number of complexities around care delivery.  In the section on formal supports it states that he NDIS does not fund case management or care coordination - but there is the availability of support coordination - but you have to know that you need this and ask for it to be in your plan (or find a planner who will suggest this).  As the paper points out, the defunding of schemes like Partners in Recovery has been very detrimental to some people.  

The paper ends a little abruptly after the findings.  It would be good to include a discussion section where this pulls together the different insights that do exist in the literature and which are scattered through the findings in terms of how people with SPMI can be better supported and the different interventions that might support this. You might also reflect on the fact that many of the issues you identify are similarly important for others who draw on a range of mainstream and NDIS services.  For example, for young people in relation to education, for formerly incarcerated people and justice services.  The issues you point out are really important and go beyond just this group to others accessing the scheme also.  

Typo line 583 - GP’s should be GPs

Author Response

Thank you very much for your helpful feedback; suggestions for improvement have been incorporated into the scoping review.

  1. It is true that there is limited consideration given to psychosocial disability and a relatively small proportion of people within the scheme with psychosocial disability. But there is also likely a much greater number of people within the scheme who have psychosocial disability as a secondary disability. It might be good to see some consideration given to this issue. 

    Consideration of secondary psychosocial disabilities has now been included in the conclusion.

  1. The methodology is mostly solid but there is no information provided regarding how the papers were coded or analysed and this needs to be added to the publication before it can be published.

    Additional information regarding coding and analysis has now been provided in 'Results' on page 7

  1. I agree that advocacy is an important issue for many NDIS participants.While support coordination is available for all participants, a number of studies have demonstrated that for some groups they need specialist support coordination delivered by individuals who clearly understand the issues facing individuals.  It appears to me that this is such a group given the number of complexities around care delivery. In the section on formal supports it states that the NDIS does not fund case management or care coordination - but there is the availability of support coordination - but you have to know that you need this and ask for it to be in your plan (or find a planner who will suggest this)

    Comment has now been made regarding lack of clarity around specialist support coordination, how to obtain it, how many participants have it in plans and how many hours funded – in formal supports NDIS  Page 12 - Lines 440-446.

  1. The paper ends a little abruptly after the findings.It would be good to include a discussion section where this pulls together the different insights that do exist in the literature and which are scattered through the findings in terms of how people with SPMI can be better supported and the different interventions that might support this. 

    Included additional insights into the conclusion and the reflection suggested at point 5 below. 
  2. You might also reflect on the fact that many of the issues you identify are similarly important for others who draw on a range of mainstream and NDIS services. 

    Reflection included within Conclusion, lines 708 & 709.

  1. Typo line 583 - GP’s should be GPs

    This has been corrected now on line 599

Reviewer 2 Report

Congratulations to the authors on a very comprehensive  review of the current state of research on the NDIS. The findings give strong indication of the research gaps, and iterate many of the concerns discussed anecdotally by service users and providers.

A few suggestions to improve the 'digestability' of this scoping review:

Please include a PRISMA-ScR checklist as an appendix to further justify the use of  a scoping review in this context.

Fig 3 - 'resnip/recapture' this image without 'Show and hide' displayed.

Results: please reiterate ( or at least summarise) the 3 research questions in the first Results paragraph to remind readers what questions these themes are addressing.

As the five themes and subthemes are extensive, a figure, map or table to summarise them would be helpful for readers to quickly grasp the findings and their relevance to the research questions.

I believe that with these additions this will be a useful paper for researchers in these health fields, and a good example of how to conduct a scoping review (and when it is appropriate).

Author Response

Thank you very much for your helpful feedback; suggestions for improvement have been incorporated into the scoping review.

  1. A PRISMA-ScR checklist has been included in Appendix B.
  2. Fig 3 - 'resnip/recapture' this image without 'Show and hide' displayed. This has been resnipped and uploaded without the 'Show and hide. 
  3. Results, the three research questions have been included in the first Results paragraph
  4. The five themes and subthemes have been summarised in a table in the first paragraph of the results section.

Reviewer 3 Report

This is a most timely and well executed scoping review on the issue of the way Australia's NDIS scheme caters to end of life care for people with serious mental illness.

The search design was sophisticated and rigorous, and its method is well described.  The research literature captured is exhaustive and the presentation of it is both analytic in its themes and very precise and insightful in drawing out its findings.

The paper makes an original and significant contribution to the literature and I recommend publication, subject to the authors checking ALL citations for any other citation flaws apart from: [12] (caps incorrectly used for title of paper); [14] (weird cite combines fields from other cites?); [52] (no cap for 'of' in journal title);  [60] (the first author's postnominal is cited as family name and the title of the paper is incorrectly in caps) and [119] (it is an Act and the reference manager has mangled 'Australian  government'  as 'Australia G'); [124] (the place of publication of journal should not be included)   

Author Response

Thank you for your feedback. All listed references have been corrected, the full reference list rechecked, and further amendments made.